# Target Recognition of Coal and Gangue Based on Improved YOLOv5s and Spectral Technology

**DOI:** 10.3390/s23104911

**Published:** 2023-05-19

**Authors:** Pengcheng Yan, Xuyue Kan, Heng Zhang, Xiaofei Zhang, Fengxiang Chen, Xinyue Li

**Affiliations:** 1State Key Laboratory of Mining Response and Disaster Prevention and Control in Deep Coal Mine, Anhui University of Science and Technology, Huainan 232001, China; pcyan1988@126.com; 2School of Electrical and Information Engineering, Anhui University of Science and Technology, Huainan 232001, China; 18168050921@163.com (H.Z.); zxf19971012@163.com (X.Z.); fxchen96@163.com (F.C.); 2021302993@aust.edu.cn (X.L.); 3Collaborative Innovation Center of Mine Intelligent Equipment and Technology, Anhui University of Science and Technology, Huainan 232001, China

**Keywords:** multispectral, improved YOLOv5s, identification and detection of coal gangue, Gaussian filtering, non-local average noise reduction

## Abstract

Aiming at the problems of long detection time and low detection accuracy in the existing coal gangue recognition, this paper proposes a method to collect the multispectral images of coal gangue using spectral technology and match with the improved YOLOv5s (You Only Look Once Version-5s) neural network model to apply it to coal gangue target recognition and detection, which can effectively reduce the detection time and improve the detection accuracy and recognition effect of coal gangue. In order to take the coverage area, center point distance and aspect ratio into account at the same time, the improved YOLOv5s neural network replaces the original GIou Loss loss function with CIou Loss loss function. At the same time, DIou NMS replaces the original NMS, which can effectively detect overlapping targets and small targets. In the experiment, 490 sets of multispectral data were obtained through the multispectral data acquisition system. Using the random forest algorithm and the correlation analysis of bands, the spectral images of the sixth, twelfth and eighteenth bands from twenty-five bands were selected to form a pseudo RGB image. A total of 974 original sample images of coal and gangue were obtained. Through two image noise reduction methods, namely, Gaussian filtering algorithm and non-local average noise reduction, 1948 images of coal gangue were obtained after preprocessing the dataset. This was divided into a training set and test set according to an 8:2 ratio and trained in the original YOLOv5s neural network, improved YOLOv5s neural network and SSD neural network. By identifying and detecting the three neural network models obtained after training, the results can be obtained, the loss value of the improved YOLOv5s neural network model is smaller than the original YOLOv5s neural network and SSD neural network, the recall rate is closer to 1 than the original YOLOv5s neural network and SSD neural network, the detection time is the shortest, the recall rate is 100% and the average detection accuracy of coal and gangue is the highest. The average precision of the training set is increased to 0.995, which shows that the improved YOLOv5s neural network has a better effect on the detection and recognition of coal gangue. The detection accuracy of the improved YOLOv5s neural network model test set is increased from 0.73 to 0.98, and all overlapping targets can also be accurately detected without false detection or missed detection. At the same time, the size of the improved YOLOv5s neural network model after training is reduced by 0.8 MB, which is conducive to hardware transplantation.

## 1. Introduction

The current energy structure in China is characterized by rich coal, poor oil and low gas content. The use of clean energy is still in the development stage, and coal still dominates the energy use [1,2,3]. The current foothold and primary task of energy transformation and development are the clean and efficient development and utilization of coal [4,5]. Coal and gangue sorting plays an important role in improving the quality of raw coal and avoiding the harm of gangue incineration on the environment [6]. Traditional methods for identifying coal gangue include manual selection [7], heavy medium method [8], wet coal selection [9], ray identification method [10,11], hardness identification method [12,13], etc. Among them, manual selection is determined by the experience of the staff, with significant errors; the heavy medium method and wet method consume a large amount of water resources, which is not conducive to large-scale promotion in arid areas; the radiant intensity of ray identification is large, which is not conducive to the long-term work of staff; the hardness identification method requires high crushing strength and dust to cause significant environmental pollution damage.

The traditional coal gangue recognition methods mentioned above have many shortcomings. Currently, machine vision technology and spectroscopy technology, as emerging technologies, have been applied in coal gangue recognition. Gao et al. [14] proposed coal gangue recognition based on deep learning, using the MobileNet target module and CBAM attention mechanism module to improve the recognition accuracy of coal gangue. Guo et al. [15] proposed a coal gangue recognition method based on a TW-RN-optimized CNN, which improves recognition accuracy while reducing recognition time. Li et al. [16] proposed to solve the problem of coal gangue recognition in visible near-infrared spectroscopy by introducing the XGBoost algorithm. Ding et al. [17] proposed a near-infrared spectral data preprocessing method for coal gangue recognition, which improves the recognition rate of coal gangue through different spectral data preprocessing methods. However, there are problems with low recognition and detection accuracy, long detection time and high rates of false or missed detections.

This article aims to improve the accuracy of coal gangue detection, reduce detection time and error rate and better apply it to industrial coal gangue separation technology. The current research has low detection accuracy and slow speed, which cannot meet the needs of industrial production well. In this paper, spectral technology is used to obtain the multispectral image of coal gangue. Through Gaussian filter and non-local average noise reduction processing, based on a YOLOv5s network, CIou Loss loss function is used to replace the original GIou Loss loss function to enhance the detection accuracy. After replacing the original NMS with DIou NMS, a new improved YOLOv5s network model was trained. The field of coal gangue recognition requires high recognition accuracy and speed, while the YOLOv5s algorithm has strong robustness and high recognition accuracy, which can accurately identify coal gangue and ensure sorting efficiency; at the same time, the YOLOv5s network model has the smallest size, strong practicality and the fastest speed, making it convenient for hardware installation and use. Other neural network algorithms have poor robustness, and the network model is too large, which is not conducive to hardware configuration and use. Comparative experiments have shown that the improved YOLOv5s can effectively detect overlapping and small targets, enhance detection accuracy, reduce detection time and reduce false-detection rates.

## 2. Introduction to YOLOv5s Algorithm

An overall block diagram of the YOLOv5s neural network algorithm is shown in Figure 1. They are the input terminal, backbone network, neck network, and prediction output terminal [18,19]. The input end includes three parts: Mosaic data enhancement, adaptive anchor box calculation and adaptive image scaling; the backbone network is mainly a convolutional neural network that aggregates and merges image features at different fine-grained levels, mainly used in some high-performance classifiers; located in the middle of the reference network and the head network is the neck network, which further enhances the diversity of target features and enhances robustness; the prediction output terminal is mainly used to output target detection and recognition results.

## 3. Network Optimization

### 3.1. Optimization of Loss Function of Intersection Union Ratio

Different choices of loss function have different effects on the output results of target detection [20]. Compared with the original GIoU Loss function of the YOLOv5 network, this paper selects the CIoU Loss function with better performance to achieve network optimization.

#### 3.1.1. GIoU Loss Function

GIoU_Loss in YOLOv5 is the loss function of the bounding box, as shown in Figure 2. On the basis of *IoU*, *GIoU* adds the area of the smallest rectangular box (dotted line box in the figure) surrounding rectangular box A and rectangular box B. The specific loss function is:(1)$$\eta_{GIoU}=\eta_{I}{}_{oU}-\frac{S_{3}-S_{2}}{S_{3}}$$

In Equation (1), $\eta_{I}{}_{oU}$ represents the size of the overlapping part. S_2_ is the area of the merging area of rectangular boxes A and B (i.e., blue + red + green area), and S_3_ is the area of the smallest rectangular box surrounding A and B (i.e., dashed box area). The loss function of YOLOv5 is:(2)$$L_{GIoU}=1-\eta_{GIoU}$$

#### 3.1.2. DIoU Loss Function

The basic principle of GIoU is based on area measurement, without taking into account the distance between the A and B rectangular boxes. In order to make the training more stable and converge faster, DIoU is proposed, which measures the distance *ρ* between the center points of rectangular boxes A and B. The diagonal length c of the bounding rectangle (dashed box) is directly taken into account, as shown in Figure 3. *ρ* represents the distance between the center points of rectangular boxes A and B; C represents the diagonal distance of the minimum closure region containing rectangular boxes A and B.

#### 3.1.3. CIoU Loss Function

The DIoU loss function does not take into account the aspect ratio of A and B rectangular boxes. In order to further improve the stability and convergence speed of training, CIoU is proposed on the basis of DIoU. The formula is as follows:(3)$$\eta_{CIoU}=\eta_{I\mathrm{oU}}-\frac{\rho^{2}(L,P)}{C^{2}}-\alpha v$$

The complete CIoU loss function is defined as:(4)$$L_{CIoU}=1-\eta_{IoU}+\frac{\rho^{2}(L,P)}{C^{2}}+\alpha v$$
(5)$$\alpha =\frac{v}{(1-\eta_{I\mathrm{oU}})+v}$$
(6)$$v=\frac{4}{\pi^{2}}{\left[\arctan\frac{w_{L}}{h_{L}}-\arctan\frac{w_{p}}{h_{p}}\right]}^{2}$$

In Equation (3), the distance between the center points of rectangular boxes A and B is determined by *ρ*. The diagonal length of the minimum bounding rectangle of A and B rectangular boxes is represented by C, and the aspect ratio similarity of A and B rectangular boxes is represented by *v*. α is a weight parameter. CIoU considers more comprehensively than DIoU, taking various situations into account, making the target box regression more stable and predicting targets more accurately. To sum up, this paper selects the CIoU Loss function with better performance.

### 3.2. Optimization of Non-Maximum Suppression

When two objects are too close, the prediction box of the other object is likely to be filtered out. Non-maximum suppression operations can enhance the selection of the target box to determine the detection box with the highest detection value. The YOLOv5 original network used weighted NMS, and in this article, DIoU-NMS is used instead of weighted NMS. The IoU indicator is used to suppress redundant detection boxes, but it can also cause erroneous suppression due to overlapping issues in the detected images. DIoU-NMS takes DIoU as the criterion for NMS, and DIoU-NMS can effectively calculate the overlapping area and the distance between the center points of two prediction boxes, which, to some extent, improves the detection of close objects. The formula is as follows:(7)$$s_{i}=\left\{ \begin{array}{l} s_{i},IoU-R_{DIoU}(M,B_{i}<\varepsilon \\ 0,IoU-R_{DIoU}(M,B_{i})\geq \varepsilon \end{array}\right.$$

In Equation (7), the score of the prediction box is represented by *S_i_*, the highest score of the prediction box is represented by *M* and the scores of other prediction boxes are represented by *B_i_. ε* represents the NMS threshold. The use of DIoU-NMS can, to some extent, improve the detection of close objects, improve the recognition of overlapping occluded targets and avoid false detections or missed detections caused by coal or gangue being too close together.

## 4. Processing and Production of Coal Gangue Dataset

### 4.1. Equipment and Data Collection

We then built a multispectral acquisition system that mainly included light sources, filters, lenses, spectral imagers and computers. As shown in Figure 4, the dashed box shows the MQ0HG-IM-SM5X5-NIR spectral imager produced by XIMEA company, with a total of 25 bands. The wavelengths from bands 1 to 25 are 891.45, 900.85, 882.80, 872.98, 959.99, 798.32, 811.15, 787.09, 773.76, 682.93, 748.57, 761.98, 736.23, 722.23, 697.47, 932.22, 939.43, 924.22, 915.28, 953.92, 851.70, 863.00, 841.41, 829.84 and 946.40 nm. Each band can achieve a resolution of 409 pixels × 216 pixels spectral image. Select LS-LHA as the light source; the filtration device is manufactured by Edmund Optics. The exposure time for spectral imager acquisition is 16.007 ms. The steps for collecting multispectral data include: initializing the multispectral image acquisition system; the multispectral imager is controlled by the software HSImager to obtain multispectral data; equipment parameter settings; data collection and organization. Figure 5 is a flowchart of coal gangue data acquisition and processing. First, we used the multispectral data acquisition system to obtain the multispectral image, and then used the random forest algorithm to select the bands of the obtained multispectral image. The RGB image is composed of three bands that are most suitable for image recognition training. The obtained RGB image is denoised via a Gaussian filter and non-local average denoising. The denoised image is the coal gangue dataset required for training.

### 4.2. Multi-Spectral Image

Using a multispectral acquisition system to obtain multispectral images of coal and gangue, it can be seen from Figure 6 that the images of coal or gangue corresponding to each band are different. It is not possible to directly distinguish which band is more suitable for identifying coal and gangue. Therefore, first, we identified the coal gangue in each band, retained the bands with a high recognition rate, selected the three bands with the lowest correlation coefficient in the reserved bands and formed pseudo RGB images for dataset production and target recognition.

### 4.3. Band Selection

The random forest (RF) algorithm is an integrated algorithm composed of multiple decision trees. RF can randomly select decision tree nodes for partitioning, which can efficiently train models with high prediction accuracy, strong noise resistance and generalization ability.

The RF principle is to give training sample set B, conduct M group random put back sampling for sample set B to obtain M group sub training set, repeat the operation for F times to obtain F training subsets as {B1, B2, B3... Bf}, train the training subsets to obtain corresponding F decision trees, combine the obtained decision trees into a random forest, obtain the corresponding classification results on the test set samples and, after the classification results are obtained, the final result is determined by a majority vote and a weighted vote.

In order to select a suitable pre-selected band, RF is used to identify coal gangue, and the coal and gangue in each band are identified 30 times using the RF algorithm. Five bands, 2, 6, 12, 18 and 21, are pre-selected based on the average recognition accuracy of different bands. Finally, correlation analysis is conducted on these five bands, and three bands, 6, 12 and 18, were selected to form pseudo RGB images for dataset production.

### 4.4. Dataset and Noise Reduction Processing

Multispectral imaging is little affected by visible light. A total of 490 sets of multispectral data were collected this time, and the three bands selected were used to form a pseudo RGB image. A total of 974 original sample images of coal and gangue were obtained, including 475 coal images and 499 gangue images. A major problem in coal gangue image recognition training is the interference of image noise, which can have a significant impact on target feature extraction and accumulate large errors in different operations, making the feature points of coal gangue difficult to extract and recognize. To ensure the extraction of clear image feature information, removing noise from coal gangue images through preprocessing is a prerequisite for target recognition. A total of 1948 pieces of coal and gangue were obtained via Gaussian filter and non-local average denoising, which not only denoised the original sample image but also increased the sample size of the data set. Figure 7 shows some original image data sets, in which coal is coal, gangue is gangue, number is image number and image format is jpg.

#### 4.4.1. Gaussian Filter

The main steps for Gaussian filtering are: scan each pixel in the image with a template, and then replace the value of the template center pixel with the weighted average gray value of the pixels in the neighborhood [21]. Gaussian filtering is used to denoise the sample image, which is beneficial to YOLOv5s neural network to extract image feature points, improve training accuracy, reduce training time and make the recognition effect better. Figure 8 is a comparison of before and after filtering (the left side is the original multispectral image of coal gangue, and the right side is the image of coal gangue after Gaussian filter).

#### 4.4.2. Non-Local Average Denoising

The non-local average denoising technology proposed in recent years has been widely applied in image denoising. The principle is to obtain the estimated value of the current pixel by weighted averaging the pixels in the image with similar neighborhood structures [22]. By using a non-local average denoising method to denoise the coal gangue sample image set, the feature points of coal gangue are added to facilitate neural network extraction and training. Equation (6) is the non-local average denoising formula, and Figure 9 shows a comparison of before and after denoising (the left side is the original multispectral image of coal gangue, and the right side is the non-local average denoised coal gangue image).
(8)$$u(x)=\sum_{y\in I}w(x,y)*v(y)$$

In Equation (8), u represents the denoised image, v represents the noisy image, x and y are the centers of two gray small windows and w(x,y) is the weight value.

## 5. Results and Discussion

This article used an SSD neural network, original YOLOv5s neural network and improved YOLOv5s neural network to train the preprocessed dataset 250 times, and we obtained their respective training weight files for recognition testing. During the training process, the red-label box represented coal and the pink-label box represented gangue, as shown in Figure 10. The numbers on the box represented a recognition accuracy of 1 during the training. Table 1 shows a comparison of two image denoising methods, indicating that the non-local average denoising method is more optimal. Table 2 shows the comparison results of different recognition algorithms after training. From Table 2, it can be seen that the improved YOLOv5s has an precision of 0.98, which is 13% and 19% higher than the original YOLOv5s and SSD, respectively; the improved YOLOv5s has a recall rate of 0.98, which is 8% and 18% higher than the original YOLOv5s and SSD, respectively; the improved YOLOv5s mAP (Mean Average Precision) reaches 0.96, which is 10% and 16% higher than the original YOLOv5s and SSD, respectively; the size of the improved YOLOv5s model is 13.6 MB, which is reduced by 0.8 MB and 64.5 MB compared to the original YOLOv5s and SSD, respectively; improving the detection speed of YOLOv5s is the fastest. It can be seen that the improved YOLOv5s outperforms SSD and the original YOLOv5s in terms of detection time, accuracy recall and average accuracy, meeting industrial requirements.

### 5.1. Ablation Experiment

This article verifies the effectiveness of CIoU Loss and DIoU-NMS on the same dataset through ablation experiments, as shown in Table 3. After replacing with CIoU Loss, precision and mAP are improved; after replacing with DIoU-NMS, the precision, recall and mAP all improved, indicating that they can effectively improve the detection accuracy of the neural network model. The combination of CIoU Loss and DIoU-NMS in the backbone network significantly improves precision, recall and mAP, effectively improving the detection accuracy of the network model for target recognition and improving the detection stability of the model.

### 5.2. Comparison Diagram of Loss Function

To compare and analyze the loss values of the three trained neural networks, we found that the smaller the loss value, the more accurate the detection frame and the higher the detection accuracy. As shown in Figure 11, the loss values of the SSD neural network, the improved network loss value (CIoU_Loss) and the original network loss value (GIoU_Loss) all decrease with the increase in iteration times. However, the improved YOLOv5s network loss value is significantly lower than the original YOLOv5s network and SSD network, indicating that the improved neural network is better than the original YOLOv5s network and SSD network.

### 5.3. Comparison Chart of Recall Rate

The recall rate represents the selection of true-positive examples from the test set by the two classifiers based on the actual results. The closer the value of the recall rate is to 1, the better the trained network model. Comparing the recall rate data obtained from each of the three neural network training processes, as shown in Figure 12, the improved YOLOv5s network gradually stabilizes and approaches 1 with the increase in iteration training times, and the effect is significantly better than the original network and SSD network.

### 5.4. Detection Result

By conducting coal gangue target recognition experiments on the trained SSD neural network, the original YOLOv5s neural network and the improved YOLOv5s neural network model, the results showed that compared to the SSD network and the original YOLOv5s neural network model, the improved YOLOv5s neural network model had the highest detection accuracy for coal blocks and gangue, with the network detection accuracy increased from 0.73 to 0.98, and there were no false detections or missed detections for overlapping targets. Meanwhile, the improved YOLOv5s neural network model has a size of only 13.6 MB. Compared to the original YOLOv5s neural network model, the size of the neural network model was reduced by 0.8 MB, making it easier to use with hardware; the average accuracy increased to 0.96, indicating that the improved YOLOv5s network has a higher accuracy in coal gangue sorting.

A comparison of coal gangue target detection results is shown in Figure 13. The recognition accuracy of the improved YOLOv5s neural network is significantly higher than that of the original YOLOv5s neural network and SSD neural network, with detection accuracy values reaching above 0.96. Part of the detection images is shown in Figure 14, with a detection accuracy of around 0.97, and there are no omissions or false detections. The experimental test results show that the optimized YOLOv5s network has better recognition performance and meets practical requirements.

## 6. Conclusions

(1) By improving the loss function of YOLOv5s, CIoU_ Loss replaces YOLOv5s with the original GIoU_ Loss loss function, which takes into account the coverage area, center point distance, aspect ratio and other conditions of prediction box and marker box, makes the loss value obtained from training smaller than that of the original network and SSD network and the network model obtained has a better detection effect.

(2) Replacing the weighted NMS of the original YOLOv5s network with DIoU-NMS, not only is the overlap area between the prediction box and the marker box taken into account but also the distance between the center points of the two prediction boxes. To improve the detection and recognition of overlapping targets, we reduced the false detection and missed detection rate of coal blocks and gangue and improved the detection accuracy.

(3) CIoU_ Loss replaces YOLOv5s with the original GIoU_ Loss function, and it replaces the weighted NMS of the original YOLOv5s network with DIoU NMS. The improved YOLOv5s neural network trained has a model accuracy of 0.98, a recall rate of 0.98, a model size of 13.6 MB, an average detection accuracy of 0.96 and a detection time of 0.019 s per frame. Compared with the original YOLOv5s network and SSD, it greatly improved the detection accuracy of coal blocks and gangue, with a short detection time, achieving the expected effect. The improved network can be applied in industrial coal preparation technology to improve the efficiency of coal gangue sorting.

(4) We utilized spectral technology to obtain multispectral data of coal gangue, and we used the RF algorithm to select three bands of images, namely 6, 12 and 18, to form a pseudo RGB image and obtain a dataset. The dataset was denoised to improve feature extraction capabilities. The improved YOLOv5s neural network obtained after training had higher recognition accuracy and better performance in target recognition testing, which is more conducive to coal gangue sorting.

## Figures and Tables

**Figure 1 sensors-23-04911-f001:**
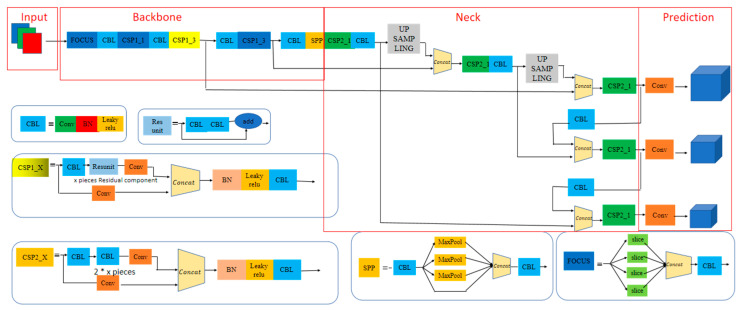
Overall block diagram of yolov5s algorithm.

**Figure 2 sensors-23-04911-f002:**
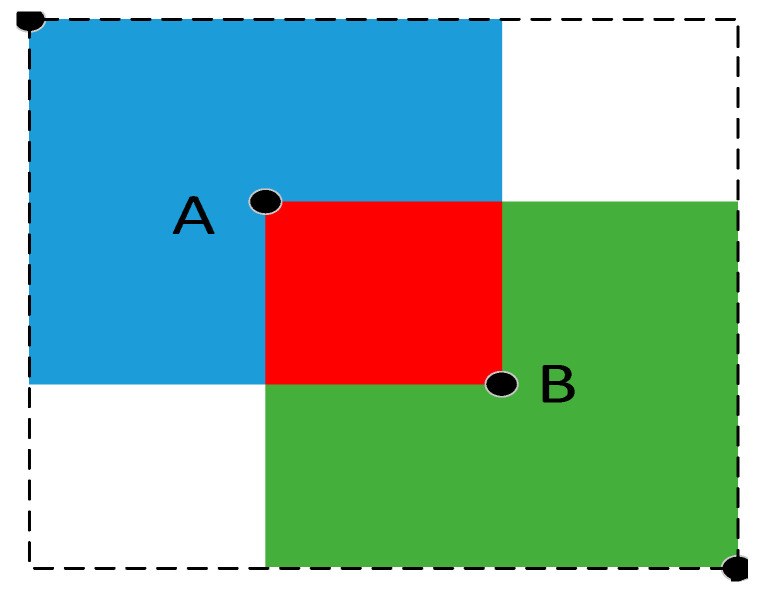
GIoU_Loss function diagram.

**Figure 3 sensors-23-04911-f003:**
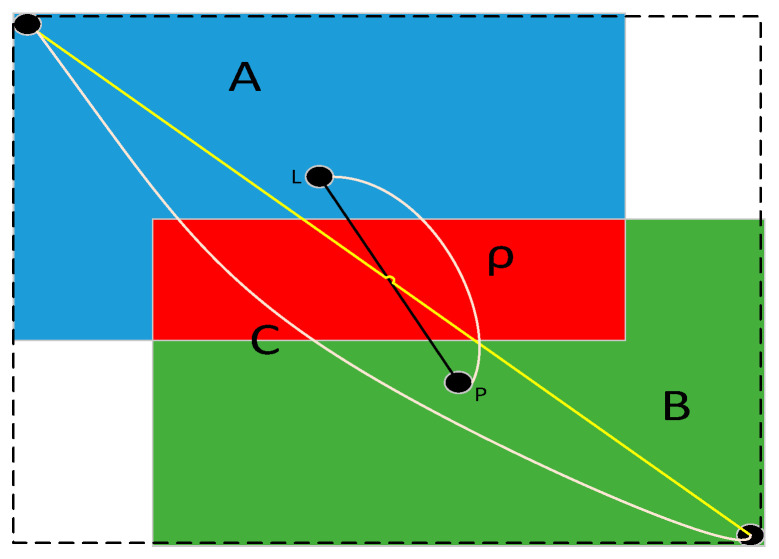
DIoU_Loss function diagram.

**Figure 4 sensors-23-04911-f004:**
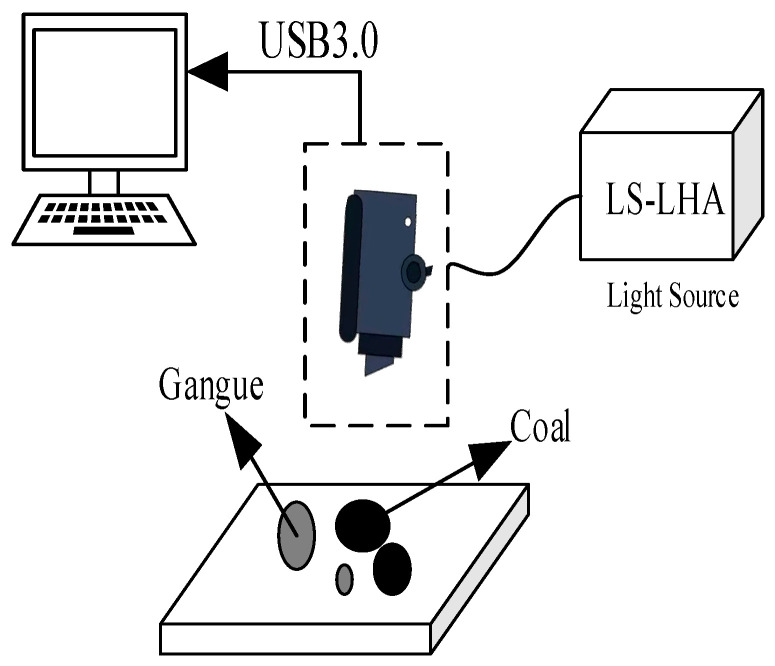
Multispectral data acquisition system.

**Figure 5 sensors-23-04911-f005:**
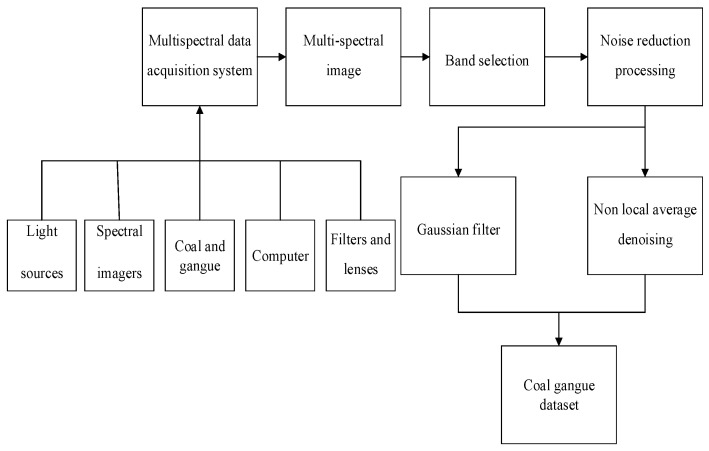
Data collection and processing flowchart.

**Figure 6 sensors-23-04911-f006:**
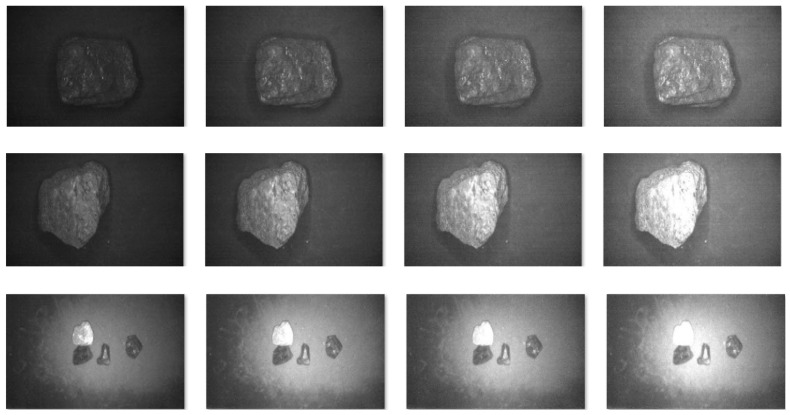
Images of coal, gangue and coal gangue mixture under different wave bands.

**Figure 7 sensors-23-04911-f007:**
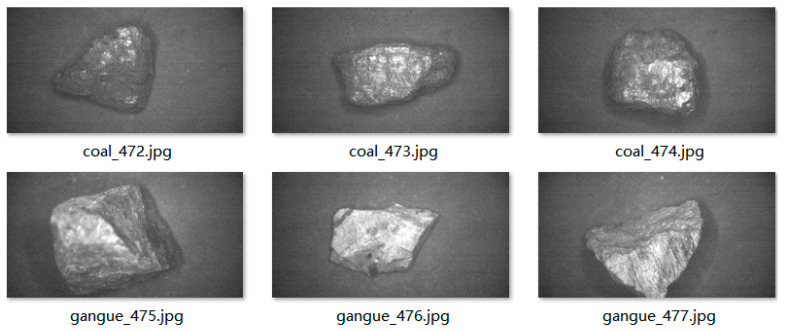
Original sample image.

**Figure 8 sensors-23-04911-f008:**
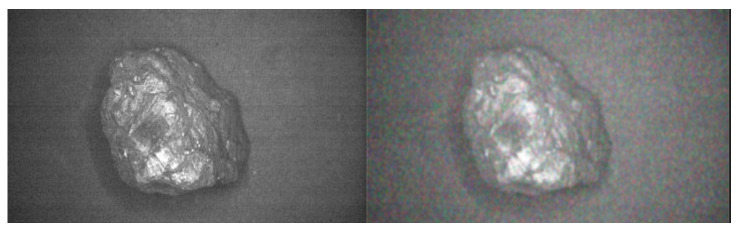
Gaussian filter comparison diagram.

**Figure 9 sensors-23-04911-f009:**
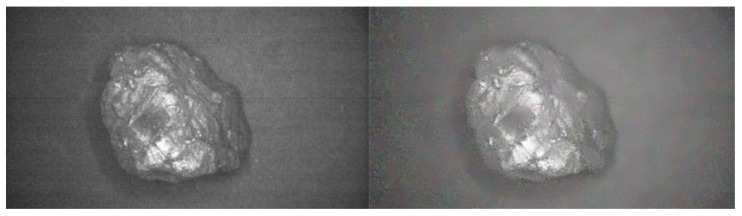
Non-local average denoising contrast map.

**Figure 10 sensors-23-04911-f010:**
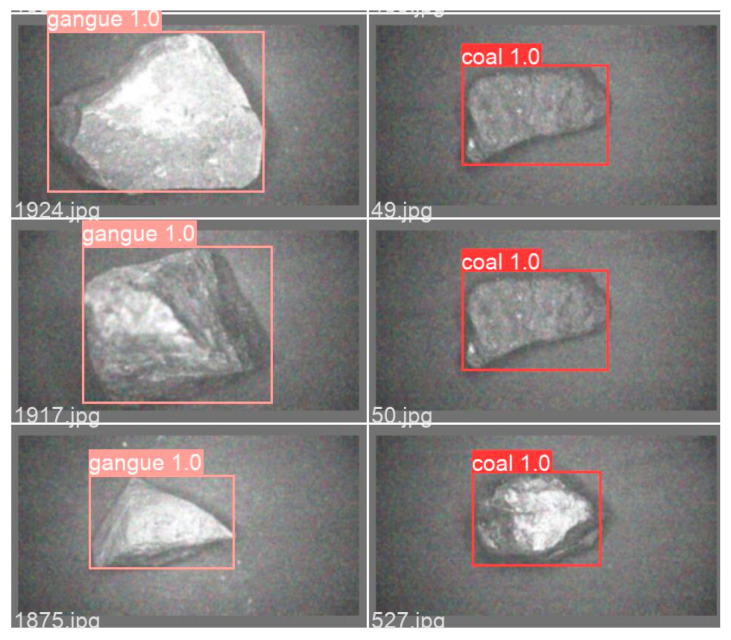
Training detection accuracy chart.

**Figure 11 sensors-23-04911-f011:**
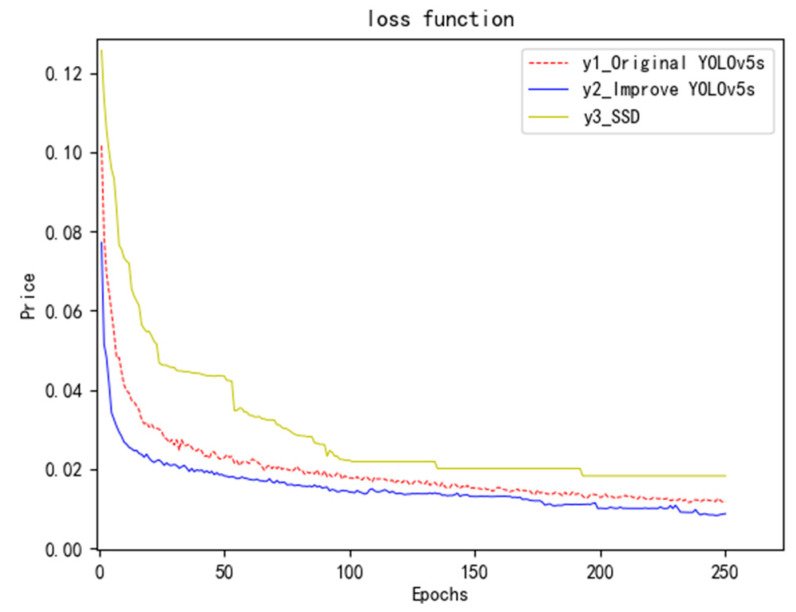
Loss function comparison chart.

**Figure 12 sensors-23-04911-f012:**
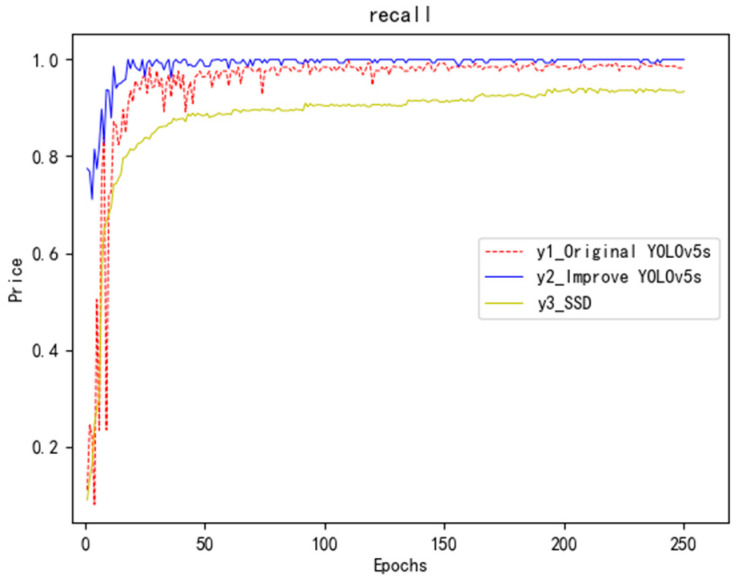
Recall ratio comparison chart.

**Figure 13 sensors-23-04911-f013:**
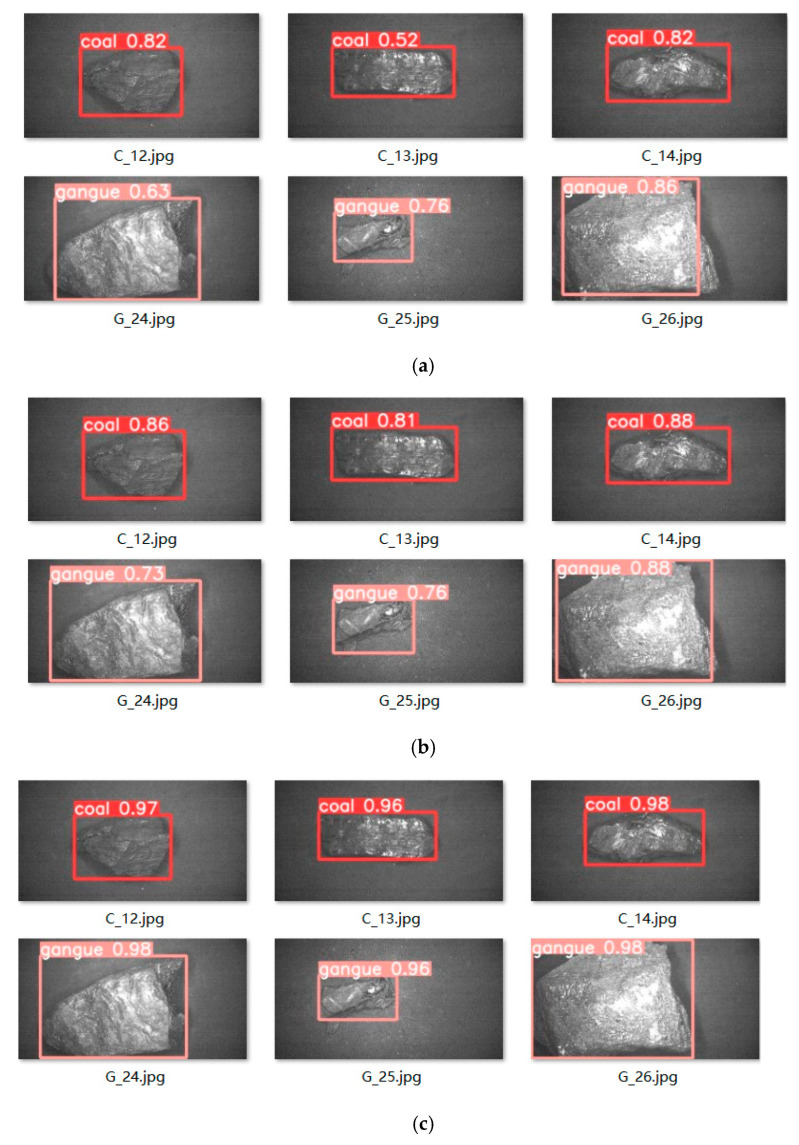
Comparison diagram of coal gangue target detection. (**a**) SSD network detection results. (**b**) Original YOLOv5s network detection results. (**c**) Improved YOLOv5s network detection results.

**Figure 14 sensors-23-04911-f014:**
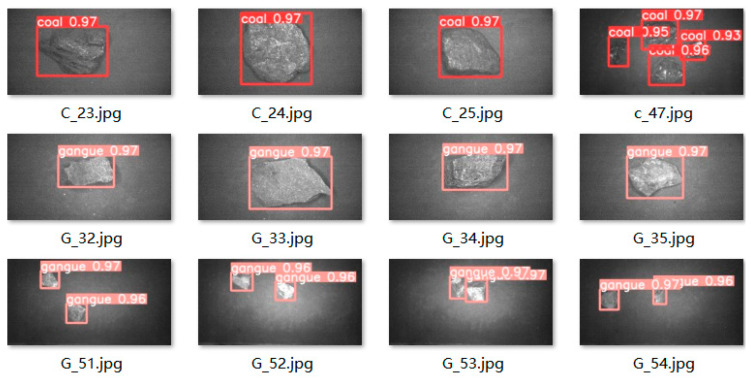
Partial coal gangue detection diagram.

**Table 1 sensors-23-04911-t001:** Comparison of image noise reduction methods.

Noise Reduction Method	Training Duration/h	mAP
Gaussian filter	8.5	0.94
Non-local average noise reduction	8.0	0.96

**Table 2 sensors-23-04911-t002:** Comparison of different recognition algorithms.

Models	Detection Time/s	Size/M	Precision	Recall	mAP
SSD	0.063	78.1	0.79	0.80	0.80
Original YOLOv5s	0.045	14.4	0.85	0.90	0.86
Improve YOLOv5s	0.019	13.6	0.98	0.98	0.96

**Table 3 sensors-23-04911-t003:** Results of ablation experiment.

YOLOv5s	**CIoU Loss**	**DIoU-NMS**	**Precision**	**Recall**	**mAP**
	**__**	**__**	0.85	0.90	0.86
	√	__	0.88	0.86	0.87
	__	√	0.89	0.92	0.88
	√	√	0.98	0.98	0.96

## Data Availability

Not applicable.

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
