# Peer review of "Target Recognition of Coal and Gangue Based on Improved YOLOv5s and Spectral Technology"

_sensors, 2023, doi:10.3390/s23104911_

Round 1

Reviewer 1 Report

A new  Target Recognition of Coal and Gangue method based on the  YOLOv5s algorithm and on the Spectral Technology is proposed.

The authors are recommended to highlight the aims and motivations of the research in the introduction and to specify the current performance limits and how the research intended to overcome them.

The block diagram of YOLOv5s in Fig 1 is unclear. The authors need to better specify the content of the block diagram in Fig. 1.

The bounding box in Fig. 2 is not a block diagram. Authors must specify its content. The same thing holds for Fig. 3.

What is the unit of measurement of the wavelengths reported in paragraph 4.1? Nanometers? Authors must specify the unit of measurement.

Section 4 should be structured better. An initial flow diagram is missing that highlights the process used and shows its components and their connections. Each component of the flowchart must correspond to one of the sub-processes described in the individual subparagraphs.

The comparative results summarized in Fig. 2 need to be described more thoroughly, especially regarding the comparisons of the performance results obtained using the original and the improved YOLOv5s.

In section 6 a description of any critical points and limitations of the proposed model and the future prospects of the research must be added.

Grammar typos are to be corrected and English must be improved.

Reviewer 2 Report

The article is of interest to the magazine. However, it needs to be improved.

1. Recently published article "Lightweight target detection for coal and gangue based on improved Yolov5s" Processes 2023, 11(4), 1268; https://doi.org/10.3390/pr11041268. It deals with the same problem. Analyze its advantages and disadvantages.

2. Subsection 3.1.3. CIoU loss function. Give details about the coefficient a in formulas (3) and (4). It is not at all clear how it is chosen.

3. Improvements using DIoU-NMS over the standard network function are not disclosed in sufficient detail.

5. Section 4.1. Wavelength is a dimensional quantity. Specify its dimensionality.

6. Section 4.1. Tell in more detail about the stages of multispectral data collection.

7. Describe in detail the experiments for which results were obtained. What is the amount of experimental research, etc.

There are some minor errors in English.

Round 2

Reviewer 1 Report

Authors have taken into account all my suggestions. I consider this paper publishable in the current form.

The quality of English is good.

Reviewer 2 Report

I am satisfied with the work of the authors to improve the article. They responded to all comments. The article may be published.